# Development of Nasal Vaccines and the Associated Challenges

**DOI:** 10.3390/pharmaceutics14101983

**Published:** 2022-09-20

**Authors:** Xuanxuan Nian, Jiayou Zhang, Shihe Huang, Kai Duan, Xinguo Li, Xiaoming Yang

**Affiliations:** 1National Engineering Technology Research Center for Combined Vaccines, Wuhan 430207, China; 2Wuhan Institute of Biological Products Co., Ltd., Wuhan 430207, China; 3China National Biotech Group Company Limited, Beijing 100029, China

**Keywords:** nasal vaccine, mucosal immunity, adjuvant, delivery system

## Abstract

Viruses, bacteria, fungi, and several other pathogenic microorganisms usually infect the host via the surface cells of respiratory mucosa. Nasal vaccination could provide a strong mucosal and systemic immunity to combat these infections. The intranasal route of vaccination offers the advantage of easy accessibility over the injection administration. Therefore, nasal immunization is considered a promising strategy for disease prevention, particularly in the case of infectious diseases of the respiratory system. The development of a nasal vaccine, particularly the strategies of adjuvant and antigens design and optimization, enabling rapid induction of protective mucosal and systemic responses against the disease. In recent times, the development of efficacious nasal vaccines with an adequate safety profile has progressed rapidly, with effective handling and overcoming of the challenges encountered during the process. In this context, the present report summarizes the most recent findings regarding the strategies used for developing nasal vaccines as an efficient alternative to conventional vaccines.

## 1. Introduction

The respiratory system in mammals performs the essential physiological function of the exchange of gases between the organism and the environment. As a consequence, the respiratory system inevitably encounters infections with various foreign pathogens [1], such as the new severe acute respiratory syndrome coronavirus (SARS-CoV-2), which is transmitted mainly via the respiratory tract, was responsible for the pandemic that began at the end of the year 2019, and has emerged as a serious threat to public health. Vaccination is one of the most effective immunological approaches currently available for enhancing the lifespan and quality of health in humans. The conventional mode of vaccination is immunization through an injection. However, this dates back to the 10th century AD in China, in which the scabs on the rashes of smallpox patients were crushed into powder and blown into the nostrils of healthy individuals for vaccination against the smallpox [2]. In the recent history of vaccine development, many clinical trials have been conducted on respiratory mucosal vaccines. The nasal route of vaccination prevents the hepatic first-pass effect and/or gastrointestinal decomposition, while only little enzymatic degradation of the vaccine antigens occurs in the nasal cavity, thereby requiring a lower dose of antigens compared to the parenteral and oral routes of immunization [3]. In addition, nasal mucosal vaccination reduces syringe usage and medical waste, due to which it is regarded as a resource-saving and environment-friendly strategy suitable for a sustainable healthcare model. Moreover, the mucosal vaccines are convenient for use as self-help vaccination, which ensures the comfort of individuals and enhances individuals compliance, thereby being suitable for mass vaccinations in the general population [4]. Since nasal mucosal vaccines have the potential to induce heightened protective immune responses at the primary site of pathogen infection, including the secretion of antibodies and tissue-resident T cell responses, these vaccines allow for blocking the epithelial cells and other cells to pathogen infection, thereby preventing the infection during the beginning stages rather than only reducing infection in later stages and preventing the further development of disease symptoms [5,6]. Meanwhile, nasal mucosal immunization has also been demonstrated to be ideal for SARS-CoV-2 vaccination [7].

## 2. Nasal Structure and Immune Responses

Several factors influence nasal vaccine design and mucosal immune responses. One of these factors is the physiological structure of the nasal cavity. The nasal cavity is divided into vestibular, respiratory, and olfactory compartments. The lining of the nasal cavity comprises the epithelium, the basement membrane, and the lamina propria, all of which are covered by a layer of tissue structure known as nasal mucosa. The human nasal mucosa has a wide surface area of 150 cm^2^, which provides a greater effective area for antigen absorption. The basal layer of the endothelium is referred to as the submucosa, which is rich in blood vessels and, therefore, facilitates the absorption of the vaccine antigens [8]. On the surface of the nasal cavity, the epithelial cells are tightly connected, forming a special physical barrier that prevents the invasion of antigenic particles, similar to the active defense against pathogenic microorganisms [9].

The mucosal immune system is the key to a host’s resistance to pathogen invasion [10]. Functionally, the nasal mucosal immune system is divided into inductive sites and effector sites, as illustrated in Figure 1. An inductive site is where the induction of the immune response begins. This is the site where the antigen first reaches the respiratory tract, then crosses the mucus layer, and is finally absorbed. The inductive site mainly comprises mucosal lymphoid follicles located within the respiratory zone, which are collectively referred to as the nasal-associated lymphoid tissue (NALT) [11,12]. NALT constitutes the Waldeyer ring in humans and can be considered to correspond to the NALT in rodents, which includes the palatine tonsils, nasopharyngeal tonsils, bilateral lingual tonsils, bilateral pharyngeal isthmus, and bilateral pharyngeal lymphatic rings [13]. NALT is the only structurally intact mucosa-associated lymphoid tissue in the upper respiratory tract. In addition, NALT is encompassed by epithelial cells and a small number of microfold (M) cells, which are invaginated to form pockets. The basal site of M cells (i.e., inside the pockets) is a region where the areas rich in B cells, T cells, macrophages, and dendritic cells (DCs) form a specialized lymphoid tissue [14,15]. M cells serve as a flexible transmembrane transporter for the non-specific as well as specific endocytosis of the antigens presented on their outer membranes. Thereafter, these antigen-containing vesicles release an extracellular secretion on the basolateral side of M cells, thereby enabling the transportation of the antigens to the antigen-presenting cells (APCs), including macrophages, DCs, and B cells, for processing and presentation [16]. In addition, certain intraepithelial or subepithelial DCs are capable of capturing antigens and migrating to the local/regional lymph nodes via the draining lymphatics [17]. The APCs that have taken up the antigen migrate to the follicular B cell zone and interfollicular T cell zone, where these cells present the antigens to the neighboring naive T cells for initiating adaptive cellular immunity [18].

An effector site in the nasal mucosal immune system is where the immune responses of B and T cells occur. The effector site includes the lamina propria (LP) and the intraepithelial layer of the respiratory mucosa [19] and is connected to the inductive site mainly through lymphocyte homing. The majority of the immune cells migrate to the inductive site to exert their effects, allowing the mucosal immunity to be relatively independent of systemic immunity and, therefore, exhibiting relative limitation [20]. This functional connection system, comprising the mucosal induction sites migrating to effector sites, is referred to as the common mucosal immune system (CMIS) [21]. CMIS may be driven by specific integrin and chemokine receptor programs [22,23]. In addition, the immune cells enter other mucosal sites and exert their effector functions, thereby linking the immune responses in different mucosal sites [24]. Consequently, antigen-specific immunity is not restricted to only the induction site and may occur even in distant mucosal zones [25]. Moreover, the antigen-activated immune cells may migrate via the bloodstream in the body and participate in systemic immune responses, thereby inducing mucosal and systemic immune responses, which are characterized by the production of IgA and IgG, respectively [6]. The secretion of local IgA antibodies block the binding of pathogens to nasal epithelial receptors [26,27]. The systemic and mucosal immune responses elicit potent humoral and cellular immune responses that may provide cross-protection [28,29].

Nasal mucosal vaccines function by inducing local tissue-resident memory (TRM) T cells, which form an essential part of mucosal surface immunity. Studies have suggested that the nasal mucosal immunization with live-attenuated influenza vaccine (LAIV) produces lung tissue-specific CD4^+^ TRM T cells and viral CD8^+^ TRM T cells with a phenotype similar to those produced upon infection with the influenza virus. On the contrary, intranasal immunization with inactivated influenza vaccines (IIV) or intramuscular immunization with LAIV reportedly could not elicit T cell responses or provide protection against viral infection. These findings suggested that both respiratory tract targeting and live-attenuated strains were required for the induction of TRM T cells [5]. Recent studies in mouse models demonstrated that while TRM cells migrate from the lung to the mediastinal lymph nodes during infection, a process referred to as “retrograde migration”, the TRM cell phenotype is maintained and provides long-term protection [30]. The T-helper cells induced after infection with the influenza virus are considered critical for the subsequent vaccination-based induction of long-term cellular and humoral immunity in the respiratory tract [31,32], further corroborating the effectiveness of mucosal vaccines.

A thorough understanding of the interactions between the structure of nasal mucosa and nasal mucosal immunity would provide valuable insights into the design of mucosal vaccine. In particular, a comprehensive exploration of mucosal APCs, innate lymphocytes, and TRM cells at mucosal sites could reveal attractive targets for vaccine design. These insights would assist in designing rational nasal vaccines against various infectious viruses (including SARS-CoV-2) and cancers using novel vaccine technologies.

## 3. Insights from the Currently Available Licensed Influenza Nasal Vaccine

Various types of injectable vaccines (including adjuvant inactivated vaccines, subunit antigen, and RNA vaccines) are currently available in the market; however, these vaccines have not yet been licensed as nasal vaccines. The only approved intranasal trivalent LAIV is the FluMist^®^ (Fluenz^®^), which has been in clinical use in the United States since 2003 [33]. In February 2012, the FDA approved FluMist^®^ Quadrivalent for people aged between 2 and 49 years [34], which was indicative of the completion of its transition from a trivalent to a quadrivalent nasal LAIV [35]. The genetically engineered LAIV virus in FluMist^®^ is a cold-adapted (ca), temperature-sensitive (ts), and attenuated (att) influenza virus. This virus is capable of replicating at 25 °C and not at 37 °C, suggesting that the LAIV form of this virus could only infect and replicate in the nasopharyngeal mucosal cells, which enhances antigen capture and delivery, thereby inducing a pattern of immunity similar to the one induced upon natural infection [34,36].

The immune response elicited by FluMist^®^ and its demonstrated safety are the main reasons for its success as a vaccine. Currently, nasal immunization with LAIV is an important mode of influenza vaccination [37,38]. In the last few decades, various other intranasal vaccine candidates against a range of human diseases have been conducted. For instance, the nasal immunization vaccine against *Bordetella pertussis* has successfully completed phase I clinical trial and entered phase II clinical trial [39,40]. Promising preclinical data have been generated for nasal immunization with Bacillus Calmette-Guérin (BCG) vaccine for the treatment of mycobacterium tuberculosis [41]. Chimeric RSV/PIV3 vaccine candidates (respiratory syncytial virus (RSV) attachment (G) and fusion (F) genes chimeric to the attenuated parainfluenza virus type 3 (PIV3)) or recombinant bivalent subunit vaccines (containing the RSV F protein and PIV3 haemagglutinin-neuraminidase (HN) protein and combined with adjuvant) have demonstrated effectiveness as nasal immunization vaccines in animal models [42,43]. Nine SARS-CoV-2 nasal vaccines, including the inactivated, live attenuated, protein subunit, nucleic acid, viral vector, and other vaccines, are currently in clinical trials, which further corroborates the feasibility of nasal vaccines [44].

## 4. Challenges of Nasal Vaccine

In the development of inactivated virus nasal vaccine, split nasal vaccine, subunit nasal vaccine, peptide nasal vaccine, and nucleic acid nasal vaccine, many factors limit the antigen absorption and bioavailability. The lipophilicity and size of vaccine molecules are important in nasal permeation. Lipophilic molecules usually diffuse via transcellular pathways, while hydrophilic molecules, particularly those with a high molecular weight, have low membrane permeability [45]. Nasal volume is another limiting factor that restricts the feasible volume for nasal administration to 25–250 µL [46]. Nasal mucosal cilia clearance and mucus barrier obstruct the antigen uptake of the vaccine, the enzymatic environment, and the local pH of the nasal mucosa, all of which would ultimately exert a negative effect on the stability of the vaccine [6]. These obstacles prevent adequate antigen delivery and the subsequent uptake and presentation of APCs residing in the nasal cavity, thereby hindering the initiation of protective immunity [47,48].

To overcome these obstacles, various strategies have been proposed to improve the efficacy of nasal vaccine in the last few years. Different antigen formulations have been developed, such as mucoadhesives (chitosan), to reduce the effects of nasal mucosal cilia on antigen clearance [49,50]. Sodium alginate (SA) and carboxymethylcellulose-high molecular weight (CMC-HMW) are reported to prolong the retention of antigens at the mucosal sites [51]. Phosphatidylcholine and cell-penetrating peptides (CPP) promote the uptake of antigens at mucosal sites [52,53,54]. Moreover, genetic optimization of the vaccines based on viruses, proteins/peptides, and nucleic acids and the use of delivery system (e.g., liposomes, nanoemulsions, and virosome) has enabled enhanced vaccine stability [55]. For instance, the use of immune enhancers, such as toxoid, cytokines, and Toll-like receptor (TLR) agonist adjuvants, reportedly promote antigen uptake and enhance the local immune response [47,56], while specific ligands or antibodies targeting APCs or M cells improve antigen uptake [57]. These strategies aim to stabilize the antigenic components, break the mucosal barrier, induce effective antigen presentation to the immune system, and activate mucosal immunity [47,52]. In this context, the present report describes the most recent findings in the development of nasal vaccines (Figure 2).

## 5. Mode of Mucosal Vaccination

Respiratory vaccination primarily targets the nasal or the lower respiratory tract. While nasal sprays, drops, and gels mainly act on the nasal cavity, the atomized aerosols and dry powder inhalation formulations act mainly on the lower respiratory tract. Nasal spray immunization is the most common method of nasal mucosal immunization compared to other nasal formulations [58]. To prevent infections of the upper respiratory tract, the particles of the nasal spray vaccine should be approximately 5 µm or above in size. Currently, the only approved vaccine for the respiratory tract is the nasal influenza vaccine, which is delivered directly to the nasal cavity as a large spray of particles to prevent deposition in the lower respiratory tract [59]. Few studies have compared the nasal spray immunization and nasal drop immunization. One study found that nasal drops have demonstrated greater effectiveness compared to nasal sprays in reducing the symptoms of nasal polyps [60], while the other study reported that nasal administration of midazolam is an excellent alternative for sedation in pediatric nuclear medicine and nasal sprays are superior to nasal drops [61], suggesting that the preferred routes of administration could vary with different drugs.

Small-particle aerosols (≤3 µm) are capable of entering the lower respiratory tract via aerosol inhalation and protecting against lower respiratory tract infections [62]. Inhalation of aerosols requires certain aerosol inhalation devices. For instance, vaccines for liquid measles may be formulated into aerosols using early modified atomizers, and have undergone extensive clinical trials. A systematic meta-analysis has confirmed that inhaled liquid aerosol vaccine against measles is safe, immunogenic, and generally well tolerated [63]. Aerosol vaccines against measles exhibit an immune response that is equivalent to or superior to injectable vaccines, while the antibody response of the former is relatively strong and durable using even a small dose of vaccine, thereby providing effective and durable protection against measles [63,64,65,66]. Inhaled dry powder live-attenuated measles vaccine (MVDP) showed superior storage stability compared to liquid vaccines. After nasal immunization, MVDP induces robust measles virus (MeV)-specific humoral and T cell responses in macaques, which has been reported to be safe, effective, and conducive to measles control [67]. Phase I clinical trial of MVDP has confirmed the safety and immunogenicity of this vaccine [68]. Dry-powder nasal mucosal inactivated influenza virus vaccine has also partially entered the clinic trial stage (ClinicalTrials.gov Identifier: NCT01488188). The nasal powder agent prolongs the retention of powder formulations on the nasal mucosa, potentially increasing local and systemic immune responses and providing better efficacy in protective immunity [51,69]. The above reports suggest that a mucosal vaccine may be used in various modes of immunization.

Special application techniques are required due to the high viscosity of nasal gel vaccines. Various gel polymers-based formulations have been developed, such as cationic cholesterol-based amylopectin (CHP) nanogels, which serve as antigen carriers for nasal vaccines to induce effective immune protection [70,71,72,73], such as the hydrogels constituted of branched polyethyleneimine and oxidized dextran or constituted of the polymer Gantrez^®^ AN119 and Pluronic^®^ F127 (PF127), the gels prepared from deacetylated gellan gum, chitosan [74,75,76,77]. All of these formulations enhance the immune response, demonstrating that nasal gels are a promising novel alternative for use as a nasal mucosal delivery system.

## 6. Live-Attenuated Vaccines

The success of attenuated nasal vaccines against influenza has inspired the development of other attenuated nasal vaccines. One example is the vaccine containing live-attenuated *Bordetella pertussis* strain BPZE1, which colonizes the respiratory tract and induces a robust protective immune response against *Bordetella pertussis* infection upon only a single nasal administration. In addition, it has demonstrated promising results in phase I clinical trials [40,55,78] (NCT02453048, NCT00870350). Another example is the attenuated RSV/ΔNS2/Δ1313/I1314L vaccine, with an application of genetic engineering technology, the interferon antagonist NS2 gene and the RSV polymerase gene codon 1313 were knocked out and the missense mutation I1314L was introduced, the resulting attenuated virus vaccine was temperature-sensitive, genetically stable, replication-deficient, and immunogenic in non-human primates [79]. Clinical trials demonstrated that this attenuated vaccine had an acceptable safety level and good immunogenicity in RSV-seronegative children, although further clinical trials in a large population are warranted [79]. An attenuated vaccine against SARS-CoV-2 named COVI-VAC was constructed by re-encoding a segment of the virus spike protein with a suboptimal synonymous codon pair (codon pair de-optimization) and introducing 283 silent (site) mutations. In animal models, COVI-VAC provided protection with a single nasal mucosal immunization [80]. The phase I clinical trials were about to be conducted (NCT05233826; NCT04619628).

Nasal administration of live-attenuated B. pertussis (BPZE1) by genetic inactivation or removal of three major toxins provides protection not only early in life (even at birth) in mice, but up to 1 year after immunization [81]. Another study also found that mice given intranasal BPZE1 and boosted with the acellular pertussis vaccine showed T cells and antibody responses [82]. Meanwhile, the BPZE1 also completed the phase I clinical trial [83,84]. BPZE1f3, a derivative of BPZE1 that contains both serotype 2 (Fim2) and serotype 3 (Fim3), produced significantly stronger protection against Fim3-only producing B. pertussis in mice than BPZE1 [85].

Table 1 presents the clinical status of nasal mucosal attenuated vaccines that are currently under development. In addition, the already marketed attenuated nasal vaccines against influenza, most of the other attenuated nasal vaccines under development (vaccines against RSV, parainfluenza virus, and avian pandemic influenza) are in the phase I clinical stage, and only a few are in the phase II clinical stage. Therefore, the nasal mucosal attenuated vaccines have demonstrated great potential, although further investigation is warranted prior to their availability to the general population.

## 7. Nasal Vaccine Adjuvants

The replication-competent viral vector-based nasal vaccines mimic the cell membrane fusion and replication capabilities of the virus during natural infections, thereby providing enhanced mucosal and humoral immunity [86]. The other nasal vaccines under development, including inactivated virus vaccine, split vaccine, subunit vaccine, peptide vaccine, and nucleic acid vaccine, require adjuvants to address their poor immunogenicity or improve delivery. The peptide-based vaccines require antigen optimization and adjuvant application, such as the HIV therapeutic vaccine Vacc-4x, which comprises four slightly modified HIV Gag p24 shared peptides and has to be used in combination with the Endocine adjuvant. In phase I clinical trials, Vacc-4x induced a dose-dependent vaccine-specific T cell response as well as the mucosal and systemic humoral responses (NCT01473810) [87]. The subunit and inactivated vaccines may also require adjuvants to enhance immune responses. Figure 2 presents the common nasal vaccines and adjuvants, and Table 2 presents the clinical progress of peptide, subunit, inactivated, and adenovirus-vectored nasal vaccines. Most of the vaccines under phase I/II clinical trials were the vector vaccine or the inactivated and subunit nasal vaccine with nasal adjuvants, such as the relatively potential LTh(αK), or nanoemulsion, liposomal, and type 1 interferon. There are no phase III clinical trials on nasal vaccines. Therefore, in-depth research is required on the development of a nasal vaccine.

The specific structure and immune response of the nasal mucosa render several of the vaccine adjuvants applicable to intramuscular or subcutaneous administration non-applicable to mucosal vaccines. The classification of the nasal mucosal adjuvants resembles mucosal adjuvants and comprises the following two categories: Immunostimulatory adjuvants and immune delivery system adjuvants. While immunostimulatory adjuvants, e.g., *E. coli* heat-labile enterotoxin (LT), cholera toxin (CT), TLR ligands, chitosan, and cytokines, activate the natural immunity directly, the immune delivery system adjuvants comprise viral vectors, virosomes, liposomes, nanoemulsions (nEs), virus-like particles (VLPs), immune-stimulating complexes (ISCOMs), and a variety of polymers, including poly(D,L-lactide-coglycolide) (PLGA), poloxamers, and alginate [88,89].

### 7.1. Immunostimulatory Adjuvants

The nasal mucosa is constantly exposed to an enormous number of pathogenic microorganisms. Generally, the endogenous cellular and molecular mechanisms in the mucosa downregulate the mucosal immune response to foreign antigens [90], probably to prevent inducing excessive immune response against harmless antigens that are frequently encountered by the mucosa. The physical barrier and the weaker immune response of the nasal mucosa render it difficult for most protein or nucleic acid-based vaccines to induce a robust protective antigen-specific mucosal immune response. However, the immunosuppressive activity of the mucosa may be evaded using mucosal adjuvants.

#### 7.1.1. Toxoid Adjuvants

The currently used mucosal adjuvants include cholera toxin (CT) and heat-labile enterotoxins (LT), in which the ADP-ribosyltransferase activity and the structural properties of the A subunit combined with the membrane-bound activity of the B subunit contribute to the adjuvant activity of LT and CT. It is believed that CT and LT induce mucosal immune responses by: (1) Increasing the permeability of epithelial cells and enhancing their antigen uptake; (2) enhancing the antigen presentation on different APCs; (3) regulating B cell differentiation and promoting IgA secretion; (4) regulating T cell proliferation and cytokine production [56,86]. However, the target cells and the underlying specific molecular mechanisms remain elusive to date.

Phosphocholine (PC) is a widely studied broad-spectrum mucosal vaccine. When conjugated with keyhole limpet hemocyanin (PC-KLH) and CT adjuvant, it induced and enhanced the IgA titer in nasal wash and saliva. In addition, it increased the response of IL-4 and IFN by CD4+ T cells. This finding indicated the feasibility of using PC combined with toxoid adjuvant as a mucosal vaccine against upper respiratory tract infections and allergic rhinitis [91]. However, nasal immunization may damage the olfactory nerve [92]. Although CT and LT have been demonstrated as powerful mucosal immune adjuvants in several studies, their application to humans is limited due to their toxicity, as evidenced in the case of CT adjuvants used in polio vaccines [93]. The generation of CT and LT mutants through genetic engineering could be one solution for eliminating or reducing their toxicity while retaining their adjuvant properties and using them in subunits or inactivated nasal vaccine. For instance, isolated LT enzyme A1 (LTA1) has demonstrated excellent efficacy and safety as an immune adjuvant for inactivated influenza vaccines [94]. LThαK lacks adenosine diphosphate (ADP)-ribosyltransferase activity and is a detoxified *E. coli* heat-unstable toxin derivative that has been studied extensively as a nasal mucosal adjuvant owing to its prolonged nasal retention and safety. The use of LThαK (NCT03784885) as an adjuvant for a trivalent inactivated influenza vaccine in phase II clinical trial revealed an acceptable level of safety and higher antigen-specific IgA responses after two nasal vaccinations [95,96]. CTA1-DD is a nontoxic mucosal adjuvant with the enzymatic properties of CT combined with B cell targeting capability. In addition, no neurotoxicity was observed after nasal vaccination with CTA1-DD, which ensures its safety as an adjuvant for nasal vaccines [97]. The HRSV fusion pre-F protein with CTA1-DD as an adjuvant could serve as a potential nasal vaccine candidate against the hRSV infection in humans [98]. LTK63, another nontoxic mutant of LT, also demonstrated the safety and efficacy of nasal mucosal adjuvants [99]. Not only the amino acid substitutions of nontoxic mutants are needed for acquiring suitable, safe, and adjuvant-active toxoid adjuvants, but also the other strategies, such as combining with other adjuvants and reducing dosage, could be considered. A similar case is the AS01 that contains the MPLA and the saponin QS-21, where the toxicity of QS-21 is reduced when combined with MPLA [100].

#### 7.1.2. Cytokine Adjuvants

The type I IFN has been used as the adjuvant for the inactivated nasal influenza vaccine in phase I clinical trial (NCT00436046), which has been successfully completed. The IL-1 family cytokines, including IL-1A, IL-1B, IL-18, and IL-33, have been used as adjuvants for the recombinant influenza virus hemagglutinin (RHA) vaccine. This vaccine-adjuvant combination could significantly stimulate the production of nasal mucosal IgA and system anti-RHA IgG after nasal vaccination in BALB/c mice [101,102]. Furthermore, DNA-IL-12 plus CTB was used as a mucosal adjuvant for DNA prime/MVA boost nasal vaccinations, which resulted in enhanced cellular systemic and mucosal genital tract immunity [103]. IL-15 has been proposed as a nasal mucosal adjuvant when developing a novel SARS-CoV-2 vaccine [104]. Other cytokines, such as those from the GM-CSF family and TNF family, IL-2, IL-4, IL-5, IL-6, IL-10, IL-12, and IL-18, are also used as mucosal adjuvants to enhance the sIgA and systemic immune responses. They were studied as candidate nasal vaccine adjuvants [105]. Overall, cytokine adjuvants were studied as mucosal adjuvants for inactivated and subunit vaccines, and preliminary research results were presented.

#### 7.1.3. TLR Agonist Adjuvants

TLRs are a group of pattern recognition receptors (PRRs), which recognize microbial pathogens and initiate host response to infection. The activation of TLRs induces a robust and immediate innate immune response, which leads to various adaptive immune responses [106]. Therefore, TLRs are ideal targets when developing effective adjuvants, especially for subunit or inactivated nasal vaccine. Monophospholipid A (MPL), the toxicity of which is 1/1000th of the toxicity of lipopolysaccharide (LPS) [107], has been used as a component of the injectable vaccine adjuvants AS01, AS02, AS04, and AS15 [108,109,110,111]. In a study, nasal administration of mice with the novel fusion protein MRPH-FIMH significantly increased the IgG and IgA contents in serum, nasal washings, vaginal washes, and urine samples, while the addition of MPL as an adjuvant further enhanced the specific humoral and cellular responses against FIMH and MRPH [112]. In another study, nasal vaccination of *Pseudomonas aeruginosa* PCRV with TLR9 agonist CpG ODN as the adjuvant significantly increased the titers of PCRV-specific IgA, which is probably a component of the disease protection mechanism [113]. The use of N3 cationic adjuvant for inactivated influenza virus vaccine significantly enhanced the systemic and mucosal-specific immune responses against influenza upon immunization, while the combination of N3 cationic adjuvant with a TLR5 agonist further enhanced these responses and durably protected against the heterologous influenza A/H1N1/CA09PDM virus [114]. The TLR3 agonist PolyI:C and the TLR7/8 agonist resiquimod (R848) are also reported to be effective in inducing mucosal immune responses [115,116,117], suggesting that TLR agonists have the potential to be adjuvants for nasal vaccines.

#### 7.1.4. STING Agonist Adjuvants

Cyclic dinucleotides that activate the stimulator of interferon genes (STING) have been evaluated as mucosal adjuvants. Nasal administration of a subunit vaccine with a synthetic cyclic dinucleotide (cyclodiguanide) adjuvant could induce protective immunity against *Mycobacterium tuberculosis* in mice, and this response was associated with the effective induction of TH17 cells [118]. Other cyclic dinucleotides, such as cyclic adenosine diphosphate (cyclic di-AMP) and cyclic adenosine diphosphate (cyclic di-GMP), have also demonstrated potential as mucosal adjuvants in previous studies [119,120]. The above reports provide a strong theoretical basis for the further development of nasal mucosal adjuvants targeting the STING pathway.

#### 7.1.5. Bioadhesive Adjuvants

Bioadhesive-based nasal vaccines are preferred for achieving nasal clearance [121]. In this context, chitosan is recognized as an established mucosal adjuvant/delivery system that has been extensively studied owing to its low toxicity, adhesion, pro-permeability, immunostimulation, ability to be absorbed in human tissues, and excellent histocompatibility with human tissues and organs [122,123,124]. In addition, the effectiveness of chitosan as a nasal adjuvant has been confirmed in a mouse model. Nasal vaccination of chitosan and pneumococcal surface protein A (PspA) reportedly induced lung PspA-specific IFNγ and STING signaling-dependent IgG1, IgG2c, and IgA responses [125]. The non-ionic block copolymer named Pluronic F127 (F127), when used with chitosan, enhanced mucosal IgA secretion [126]. Moreover, EG-coated polylactic acid 80 enhanced IgG and IgA immune responses. PEG-grafted chitosan nanoparticles reportedly enhanced the nasal absorption of insulin [127]. These studies suggested that chitosan may be used in combination with other mucosal adjuvants for protein nasal vaccine to synergistically enhance the immune effect.

Nasal vaccination of protein-coated chitosan nanoparticles encapsulating the influenza mRNA molecules reportedly protected chickens against avian influenza [128]. The nasal mucosal immunization with SC2-spike DNA vaccine transported on a modified gold-chitosan nanocarrier resulted in an immune response with high levels of antibodies (IgG, IgA, and IgM) and lung mucosal and TRM T cells [129]. Recently, chitosan-based nasal vaccines have emerged as a research hotspot, and with continued research, chitosan is expected to play a further important role as an immune adjuvant combined with protein and nucleic acid-based nasal vaccines.

Another substance that exhibits mucoadhesive properties is PLGA. PLGA and PLGA/chitosan (chit) nanoparticles (NPs) with mucoadhesive properties may be used to encapsulate ropinirole hydrochloride (RH), a dopaminergic agonist against Parkinson’s disease, to facilitate RH delivery [130]. The feasibility of mucoadhesion adjuvants for clinical use needs to be validated by further research.

#### 7.1.6. Cell-Targeted Adjuvants

M cell-targeting vaccines are based on the endocytic uptake of foreign antigens by M cells. UEA-1 is a plant lectin that is used as an M cell-specific ligand for targeting α-l-fucose (murine PP M cells). The HIV peptide and UEA-1 entrapped in PLG microparticles reportedly enhanced the mucosal and systemic immune responses through nasal immunization, which validated UEA-1 as an M cell-targeting nasal vaccine component [131]. Certain specific molecules expressed highly on the surface of M cells, such as glycoprotein 2, uromodulin, the cellular prion protein (PrPC), α(1,2)-fucose-containing carbohydrate, C5aR, α-2,3 sialic acid, and β1 integrin, are also used as receptors in M cell targeting [132,133].

To induce an immune response in the respiratory tract, the antigens must be taken up by APCs [134]. The activation, maturation, and proliferation of DCs are regulated by a range of molecules, including viral components, antigenic molecules of bacterial origin, growth factors, and cytokines [135,136]. For instance, FMS-like tyrosine kinase 3 (FLT3) is an important receptor tyrosine kinase in cell signaling. The binding between FLT3 and the ligand (FL) of FLT3 is reported to induce conformational changes in the pre-existing homodimer and consequently activate the tyrosine kinase, thereby inducing a DC response [137]. Nasal immunization of mice with ovalbumin (OVA) along with a plasmid encoding FL (pFL) as the nasal DC-targeting adjuvant reportedly induced OVA-specific SIgA and systemic IgG and IgA Ab responses [138]. The nasal administration of a combination of DNA plasmids encoding the FLT3 ligand (pFL) and CpG oligodeoxynucleotide 1826 (CpG ODN) (FL/CpG), as the nasal mucosal adjuvant, effectively enhanced the DC responses, balanced the Th1 and Th2 type cellular responses, and provided rFimA-specific IgA-based protection in the respiratory tract against *Porphyromonas gingivalis* [139].

Furthermore, the synthetic glycoside named α-galactosylceramide (also referred to as α-GalCer; composed of an α-linked sugar and a lipid fraction) binds to the non-classical MHC I molecule CD1d, thereby inducing the proliferation of natural killer T (NKT) cells and secretion of IFNγ, which results in the expression of the semi-invariant Vk14 T cell receptor and ultimately the induction of innate and adaptive immunity [140]. The optimized α-GalCer exhibited better solubility and was effective in stimulating the NKT cells, thereby inducing the release of Th1/Th2 cytokines, which significantly elevated the titers of systemic IgG and mucosal IgA antibody and enhanced the production of cytotoxic T lymphocytes in mouse and in vitro human system. These findings suggest that α-GalCer analogs with branched acyl chains could serve as effective mucosal adjuvants for inducing protective immune responses against influenza virus infection [141,142].

Activated mast cells contribute to stimulating inflammatory mediators and releasing immune cells. The mast cell activators named polymeric compound 48/80 (C48/80) and the cationic peptide mastoparan 7 (M7) have been used as adjuvants in nasal mucosal immunization [143], for example, pneumonia vaccine in combination with C48/80 could combat lethal pneumococcal infection [144]. Certain small molecule mast cell activators have also exhibited certain mucosal adjuvant effects and may be novel mucosal adjuvants.

### 7.2. Delivery System Adjuvants

The size of the delivery particles affects the cellular uptake and pharmacokinetics of the delivered molecules. In addition, the surface modification of the delivery particles alters the specificity and effectiveness of the ligand–APC interactions [145]. Viral vectors, when used in nasal vaccine, proliferate in the host tissues after nasal mucosal immunization, offering greater probability of stimulating immune responses. In the case of delivery vectors other than the virus-based system, factors such as size, biodegradability, adhesion, internalization rate, pH sensitivity, antigen release rate, and adjuvant activity have to be considered, as well.

#### 7.2.1. Viral Vectors

Different vectors usually elicit different immune responses and, therefore, provide different degrees of protection. Therefore, certain vectors may be preferred when developing nasal mucosal-targeted vaccines owing to their unique properties. For instance, adenovirus type 5 (Ad5) is a common respiratory virus, and Ad5-based recombinant vectors have been used widely as vaccine candidates against SARS-CoV-2, influenza, Ebola, HIV-1, and other infectious diseases [146,147]. Notably, nasal mucosal immunization with an Ad5 vector-based vaccine provides better mucosal immunity and protection, which can effectively reduce viral load in the upper respiratory tract. Therefore, this vaccine is a suitable candidate for preventing SARS-CoV-2 infection and transmission and mitigating the impact of the pandemic [146,148,149]. Another nasal vaccine developed against SARS-CoV-2 was based on the replication-incompetent human parainfluenza virus type 2 (HPIV2) [150]. This vaccine delivers ectopic genes as stable RNA molecules and ectopic proteins to the membrane, thereby inducing high levels of neutralizing IgG and mucosal IgA antibodies in mice for the neutralization of SARS-CoV-2 spike proteins. The vaccine is currently in clinical trial on healthy human volunteers [151]. Human parainfluenza virus types 1 and 3 (HPIV1 and HPIV3) may be modified through genetic engineering to express the RSV fusion pre-F proteins, which serve as nasal vaccine candidates with protective effects [152,153]. Similarly, the influenza virus is a promising vaccine vector, which is being used for developing a novel nasal vaccine against SARS-CoV-2. This novel vaccine induces serum neutralizing antibodies comparable to the levels generated in the natural infection [154,155]. Recombinant attenuated influenza viruses expressing the structural domain of the RSV G protein reportedly induced robust IgA-specific immune responses and TRM T cell responses in the lung and bronchoalveolar fluid of mice, thereby protecting the mice from RSV attack [156,157]. Nasal immunization with HIV vaccine plus BCG or influenza virus-based vectors was reported to promote HIV-specific cellular and humoral immune responses in the airway and vagina of mice [158]. Nasal or oral administration of the baculovirus-vectored human papillomavirus (HPV) vaccine demonstrated protective effects against vaginal HPV infection [159].

However, the immunoprotection of the Ad5 vector-based vaccines may be reduced due to the pre-existing anti-Ad5 immunity in humans resulting from natural exposure or prior vaccination. The vector-specific serum antibodies were reported to severely impede the seroconversion (change from seronegative to seropositive condition) of neutralizing antibodies against SARS-CoV-2 in volunteers with intramuscular vaccination [160]. The nasal mucosal immunization against Ad5-S-nb2 in macaques was reported to induce a significantly delayed antibody production against Ad5 and lower serum antibody titers compared to intramuscular administration [148]. This finding suggested that nasal mucosal immunization with adenoviral vector-based vaccines is less affected by pre-existing Ad5 antibodies [161,162]. Despite the scientific concerns regarding an increased risk of HIV infection upon using adenoviral vector-based vaccines [163], the HAdV-26 COVID-19 vaccine has been used widely in large-scale trials or after emergency authorizations, and no increase in the HIV-1 infection rates was reported [163]. The LAIV vector-based SARS-CoV-2 attenuated vaccine (dNS1-RBD) has completed phase I and II clinic trials, which revealed weak T cell immunity in the peripheral blood and weak humoral and mucosal immune responses against SARS-CoV-2 in the vaccinated recipients. However, further studies are warranted to validate the safety and efficacy of the nasal vaccine as an alternative immunization route to the currently used intramuscular SARS-CoV-2 vaccine [164].

In clinical trials, viral vector-based vaccines may prove to be significantly ineffective when the same vector or similar vectors are used, as the efficacy of human vaccines is reduced in individuals that have been previously infected with these viruses. Several non-homologous viral vectors have been studied extensively, among which is the Newcastle disease virus (NDV), which was used as a vaccine vector for SARS in 2003 and SARS-CoV-2 in 2019 [165,166,167]. NDV vectors expressing wild-type S or membrane-anchored S and without polybasic cleavage sites may also be used as vaccine vectors against SARS-CoV-2. These COVID-19 vaccine candidates reportedly protected mice from mouse-adapted SARS-CoV-2 attack without any detectable NDV titers and viral antigens in the lung [168]. Another example is the avian paramyxovirus type 3 (APMV3), which was used as the vaccine vector against SARS-CoV-2 [169]. In addition, the rabies virus (RV) vaccine strain, a recombinant vesicular stomatitis virus (VSV), and cowpea mosaic virus (CPMV) have been used as candidate vaccine vectors [24,170]. However, the mutations and biosafety of these candidate viral vectors remain unclear to date, warranting further investigation prior to application.

#### 7.2.2. Liposome Delivery System

Liposomes are spherical vesicles with a bilayer structure formed by the phospholipids dispersed in an aqueous solution. Liposomes have diameters ranging from 10 nm to several μm and contain an aqueous phase encapsulated inside their structures [171]. The encapsulated aqueous phase contains water-soluble compounds (e.g., proteins, peptides, and nucleic acids), while the lipophilic compounds (e.g., antigens and adjuvants) are embedded in the lipid bilayer of the liposome structures. Liposomes offer the advantages of preventing antigen degradation, delivering poorly soluble drugs, and reducing drug toxicity. In addition, the encapsulation of the drug inside a liposome allows for the customization of the localization and distribution of the drug at the target site. These advantages facilitate the effect of vaccine antigens in the nasal mucosal environment.

Cationic liposomes to be used as a vaccine adjuvant-delivery system may be prepared using 1,2-dioleoyl-3-trimethylammonium-propane (DOTAP), dimethyl dioctadecyl ammonium bromide (DDA), and dimethylaminoethane-carbamoyl-cholesterol to further enhance the immunogenicity of vaccines [171]. In comparison to non-cationic lipid vesicles, DDA- and PEG-based cationic liposomes used as a vaccine adjuvant-delivery system further enhance the immune responses [172]. In addition, nasal vaccination with cationic liposomes prepared from DOTAP and carbamoyl-cholesterol are efficiently internalized by DCs present in the NALT, which induces the production of antigen-specific IgA and T cell responses in the nasal mucosa tissues [173,174]. Nasal mucosal vaccination with pneumococcal surface protein A plus cationic liposomes composed of DOTAP and cholesteryl 3β-N-(dimethylaminoethyl)-carbamate (DC-chol) (DOTAP/DC-chol liposome) reportedly induced protective immunity against *Streptococcus pneumoniae* infection in mice [175]. CAF01 is a novel liposomal adjuvant system comprising cationic liposomal carriers (DDA and glycolipid immunomodulators (alginate 6,6-dibehenate (TDB)) with a stable structure [176]. Studies have demonstrated that nasal vaccination with CAF01-based vaccines for the prevention of influenza or *Streptococcus pyogenes* was effective in inducing the production of mucosal effector T cells and IgA immune responses and also protected the vaccinated model animals. In addition, the spleen of CAF01-vaccinated mice produced four-fold higher levels of antigen-specific IFN-γ responses compared to the mice with non-adjuvanted vaccination. This finding demonstrated that CAF01 significantly enhanced the levels of vaccine-specific serum IgG [177]. Endocine™, which consists of the lipids mono-olein and oleic acid, has been used in a nasal vaccine (Vacc-4x). The clinical trial confirmed the safety and mucosal and systemic antibody responses, as well as the dose-dependent vaccine-specific T cell responses [87]. VaxiSomeTM, a liposome composed of a novel polycationic sphingolipid complexed with cholesterol is another effective adjuvant/carrier system for nasal mucosal immunization against influenza in mice [178,179]. Furthermore, the stabilization of liposomes through the layer-by-layer deposition of polyelectrolytes is reported to increase antigen-specific IgA and IgG antibody levels and T cell-based immune responses [180]. This finding suggests that the optimization of liposomes significantly impacts their stability and vaccine-induced immune responses.

Cationic cyclodextrin-polyethylenimine 2 k conjugate (CP 2 k) complexed with anionic mRNA encoding HIV gp120 reportedly overcame the epithelial physical barrier, prolonged nasal retention, enhanced the paracellular delivery of mRNA, minimized toxin absorption in the nasal cavity, induced balanced Th1/Th2/Th17 types, and achieved a robust systemic and mucosal anti-HIV immune response [181]. Cationic cyclodextrin-PEI conjugated to the IVT mRNA encoding OVA, significantly promoted nasal mRNA vaccine delivery in mice after nasal vaccination and also stimulated DC maturation and migration, which further enhanced the humoral and cellular immune responses [182]. The advent of mRNA vaccines against SARS-CoV-2 may further facilitate the development of lipid nanoparticle nasal mucosal RNA vaccines against respiratory infectious diseases [183]. This suggests that liposomes could be used for delivering several forms of antigens.

#### 7.2.3. VLP Delivery System

VLPs are self-assembling biomolecules that closely resemble native viruses. VLPs do not contain any genetic material, which is beneficial for APC recognition and uptake and also for BCR crosslinking [184,185]. In comparison to recombinant protein vaccines, VLPs exhibit excellent stability, which is conducive to the development of mucosal vaccines [186]. For instance, VLP-based norovirus vaccine used with chitosan adjuvant has exhibited effective nasal immunization [187]. In addition, nasal immunization with Norwalk VLP was reported to induce antibody production at distal mucosal sites [188].

#### 7.2.4. Virosome-Mediated Delivery System

Virosomes are vesicles with a monolayer or a bilayer phospholipid membrane that encapsulates virus-derived proteins, although these could not replicate. Virosomes are able to fuse with target cells, and owing to their delivery effectiveness and demonstrated safety, virosomes are considered to be used for directly delivering the vaccine antigens inside the host cells [189]. Virosomes containing surface HIV-1 gp41-derived P1 lipidated peptide (MYM-V101) were reported to induce mucosal anti-gp41 antibodies against conserved gp41 motifs. In addition, these virosomes are expected to possess the HIV-1 transcytosis inhibition activity and may, therefore, contribute to the reduction in sexually transmitted HIV-1 (NCT01084343) [190]. Nasal immunization with nasal virosome-formulated influenza subunit vaccine in a ferret model prevented viral shedding almost entirely and also protected against homologous viral attack compared to parenteral immunization [191]. The results of phase II clinical trial of an influenza vaccine plus HLT and nasal virosomes as adjuvants revealed an efficient induction of IgA-neutralizing antibodies in the mucosa [192]. Moreover, Bárbara Fernandes reported the successful construction of a virosome-based COVID-19 vaccine candidate [193], which indicates good prospects for using virosomes as a vaccine delivery system.

#### 7.2.5. ISCOMs

ISCOMs are negatively charged nanoparticles with cage-like structures. ISCOMs are composed of phospholipids, cholesterol, and Quil A (the saponin derived from the bark of the *Quillaja saponaria* plant), all of which are capable of accommodating a wide range of antigens. Hydrophobic antigens may be embedded or anchored directly into the colloidal structure of the lipid, while hydrophilic antigens require certain modifications prior to their effective loading into ISCOMs [194]. Initially, ISCOMs were used for injectable vaccination. Currently, these are being gradually introduced to mucosal vaccination (e.g., nasal and oral). ISCOMs-based vaccines induce an antigen-specific mucosal IgA response, producing serum IgG antibodies and cytotoxic T cells (CTL) [195]. Similar to CT and LT, ISCOMs break the immune tolerance and exert potent mucosal adjuvant activities, secreting IgA and inducing systemic IgG immune responses and CTL responses [196].

ISCOMs-based vaccines containing influenza virus, *Mycobacterium tuberculosis*, RSV, and measles antigen have been indicated to effectively induce an immune response in the body after nasal/pulmonary vaccination, producing high titers of IgA in the nasal cavity and the lung [197,198,199,200,201,202,203]. In certain cases, this local mucosal inoculation produces greater immunity compared to that produced upon injectable vaccination. ISCOMs prepared using *Quilaja brasiliensis* in place of Quil A may also be used to encapsulate the OVA antigen and then used as a vaccine adjuvant-delivery system to induce the production of local mucosal and distal IgA secretion after nasal vaccination [203]. Furthermore, the addition of DNA plasmids to ISCOMs reportedly induced local cellular and antibody responses against *Haemophilus influenzae* after nasal immunization [204]. The combination of CTA1-DD and ISCOMs was reported to enhance the immunity against BCG through the recruitment of antigen-specific effector cells to the infected site [200].

However, the use of ISCOMs as a mucosal vaccine adjuvant-delivery system for nasal vaccination has certain limitations. The hydrophilic antigens must be modified to improve lipophilicity for effective delivery, while certain saponin-based adjuvants are highly toxic when used at high doses. These limitations have curbed the development and application of ISCOMs in vaccine formulations to a certain extent.

#### 7.2.6. NE-Based Adjuvants

Although emulsion-based adjuvants have demonstrated success in injectable vaccine applications, these adjuvants have not reached the standard of mucosal vaccine clinical application to date [205]. Nasal immunization of recombinant anthrax protein vaccine (BW-1010) in nanoemulsion (NE) adjuvant (oil-in-water emulsion) is in the currently ongoing phase I trial (NCT04148118) conducted by Bluewillow Biologics. The NE adjuvant was observed to promote Th1 cellular immunity and Th17 cellular immunity, which is consistent with the results obtained for the *Mycobacterium tuberculosis* vaccine plus NE adjuvant system [206,207]. W805EC is another oil-in-water NE adjuvant, when combined with the inactivated influenza vaccine, the protective hemagglutination inhibiting (HI) antibody and influenza-specific IgG and IgA were elicited [208]. PEG-b-PLACL (PELC) is a squalene-based oil-in-water NE adjuvant that reportedly promotes antigen penetration and uptake in the nasal mucosa and enhances protein interactions [209]. A preclinical study conducted on a guinea pig model of infection has confirmed the protection provided by this vaccine system, which was also correlated with systemic and mucosal antibody induction. NB-1008 is a surfactant-stabilized soybean oil-in-water NE adjuvant. The nasal mucosal immunization with the NB-1008 adjuvant comprising influenza virus antigens was reported to induce mucosal and serum antibody responses along with a robust cellular Th1 immune response [210]. Moreover, 5–15 nm NEs prepared using coupling techniques, such as lipopeptide coupled with polylysine nuclei, effectively promoted the systemic, mucosal, and cellular immune responses after nasal vaccination, thereby protecting against *Streptococcus pyogenes* infection [211].

## 8. Conclusions

Nasal mucosal immunization offers the advantage of inducing mucosal sIgA, cellular and systemic immune responses, and immune responses at distal mucosal sites, which would improve the effectiveness of vaccine against pathogens. The success of nasal LAIV has inspired the development of numerous other nasal attenuated vaccines, vector-based vaccines, and nasal mucosal adjuvant vaccines. The advent of genetic engineering further facilitated the success of attenuated vaccine strains, with several of the nasal mucosal attenuated vaccines currently in phase I, II, and III clinical trials, indicating good prospects for nasal mucosal attenuated vaccines. The viral vector-based vaccines result in greater stimulation of the immune response, although whether the vector-targeted immune response reduces the immunogenicity of the vaccine is still confusing. In the case of inactivated vaccines, subunits vaccines, and nucleic acid vaccines, the addition of mucosal adjuvants, such as toxoids, cytokines, and TLR agonists, enhances the immunogenicity of antigens, while ISCOMs, NEs, VLPs, and virosomes are effective in delivering antigens and ensuring antigenic stability. Some adjuvant-added nasal vaccines comprise both immune-enhancing and delivery-facilitating adjuvants to promote the delivery of vaccines to immune cells and also enhance the immunogenicity of vaccines. The establishment of novel adjuvanted nasal vaccines as safe and effective requires validation through extensive preclinical and clinical trials, and although the successful development of nasal vaccines has a long way to go, it owns positive implications. If the safety of the adjuvant-added nasal vaccines is guaranteed, the effective candidate for peptide, subunit, inactivated nasal vaccines would be enough to warrant success.

## Figures and Tables

**Figure 1 pharmaceutics-14-01983-f001:**
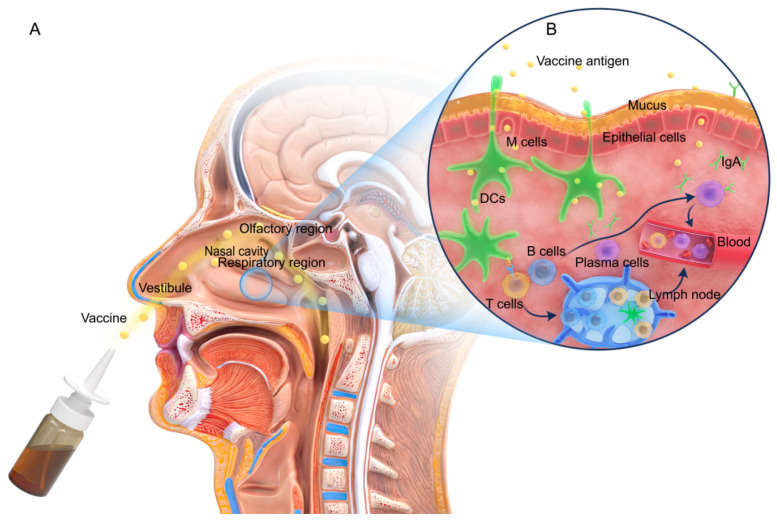
Nasal structure and the mucosal immune response. (**A**) Structure of the nasal cavity. (**B**) The immune response mechanism in the nasal mucosa. M cells or DCs (green) take up the antigens at the induction site, migrate to NALT, and interact with T cells (yellow) to activate B cells (blue) into plasma cells (purple). A few of the plasma cells migrate to the effector site and secrete IgA antibodies. In addition, the remaining B cells and the activated T cells pass through the blood, induce cellular and IgG-based systemic immune responses, and may even differentiate into memory B and T cells.

**Figure 2 pharmaceutics-14-01983-f002:**
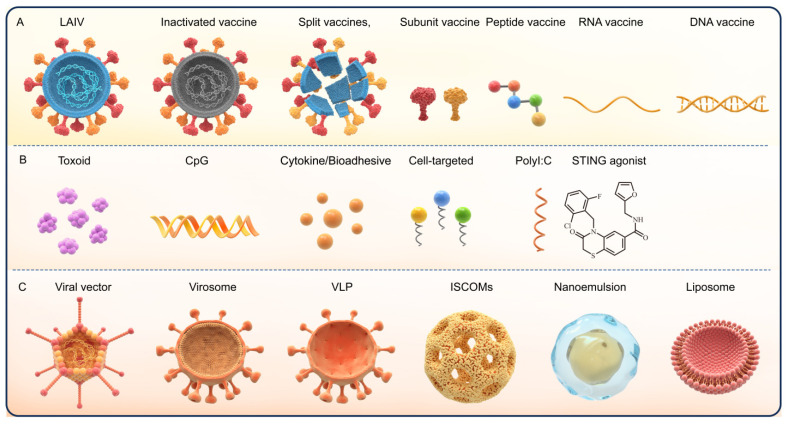
The different types of nasal vaccines and adjuvants. (**A**) The subtypes of nasal vaccines. (**B**) The subtypes of immune enhancement adjuvants. (**C**) Adjuvant delivery system.

**Table 1 pharmaceutics-14-01983-t001:** The current progress in the clinical development of nasal attenuated vaccines.

Disease	Vaccine Identification	ClinicalTrials.gov Identifier	Phase	Sponsor	Status
Pertussis	BPZE1	NCT01188512 NCT02453048	1	Inserm	Completed
	Vaccine GamLPV	NCT04036526	1/2	Gamaleya Research Institute of Epidemiology and Microbiology	Unknown
	BPZE1	NCT03541499	2	NIAID	Completed
	BPZE1	NCT03942406 NCT05116241	2	ILiAD Biotechnologies	Completed/Recruiting
COVID-19	COVI-VAC	NCT05233826 NCT04619628	1	Codagenix, Inc.	Active, not recruiting
RSV	RSV ΔNS2/Δ1313/I1314L RSV D46/NS2/N/ΔM2-2-HindIII RSV LID ΔM2-2 1030s RSV 6120/∆NS1 RSV 6120/F1/G2 RSV 6120/∆NS2/1030s RSV cps2 Vaccine RSV MEDI ΔM2-2	NCT03227029 NCT03102034 NCT02794870 NCT02952339 NCT03916185 NCT03422237 NCT03099291 NCT04520659 NCT03596801 NCT03387137 NCT01968083 NCT01852266 NCT02237209 NCT01893554 NCT01459198	1/2	NIAID	Completed/Recruiting/ Active, not recruiting
	MV-012-968	NCT04227210	1	Meissa Vaccines, Inc.	Recruiting
	RSV vaccine formulation 1	NCT04491877	2	Sanofi Pasteur, a Sanofi Company	Recruiting
Parainfluenza Virus Diseases	rHPIV1 84/del170/942A Standard Dose HPIV2 rHPIV3cp45 HPIV3/ΔHNF/EbovZ rB/HPIV3 HPIV3-EbovZ GP	NCT00641017 NCT01139437 NCT01021397 NCT03462004 NCT01254175 NCT00366782 NCT00308412 NCT02564575	1	NIAID	Completed
RSV and PIV3	MEDI-534	NCT00345670 NCT00493285 NCT00686075	1/2	MedImmune LLC	Completed
Influenza	H2N3 MO 2003/AA ca Vaccine Live-attenuated H7N9 A/Anhui/13 ca influenza virus vaccine Live Influenza A Vaccine H7N3 (6-2) AA ca H9N2 (6-2) AA ca H2N2 1960 AA ca H6N1 Teal HK 97/AA Influenza A H7N7 H5N1 (6-2) AA ca	NCT01175122 NCT01995695 NCT02251288 NCT02151344 NCT00516035 NCT01854632 NCT00853255 NCT00380237 NCT00722774 NCT00734175 NCT00922259 NCT00110279 NCT01534468 NCT00488046	1/3	NIAID	Completed
	SIIL LAIV cH8/1N1 LAIV LAIV H7N3	NCT01625689 NCT03300050 NCT01511419	1/2	PATH	Completed/Recruiting
	GHB16L2	NCT01369862	1/2	AVIR Green Hills Biotechnology AG	Completed
	CAIV-T UniFluVec	NCT00224783 NCT00192309 NCT00192413 NCT04650971	1/2/3	MedImmune LLC	Completed
	A/17/CA/2009/38 (H1N1)	NCT01666262	1/2	Mahidol University	Completed
	Lactobacillus rhamnosus	NCT00620412	1	Tufts Medical Center	Completed
Human metapneumovirus	rHMPV-Pa vaccine	NCT01255410	1	NIAID	Completed
Meissa	MV-012-968	NCT04690335	2	Meissa Vaccines, Inc.	Completed
Sendai virus	Sendai virus vaccine	NCT00186927	1	St. Jude Children’s Research Hospital	Active, not recruiting

**Table 2 pharmaceutics-14-01983-t002:** Clinical progress in the development of peptide, subunit, inactivated, and viral vector-based nasal vaccines.

Disease	Vaccine Identification	Type	Adjuvant	ClinicalTrials.gov Identifier	Phase	Sponsor	Status
Influenza	Vaccination	Subunit	Liposomal	NCT00197301	1/2	Hadassah Medical Organization	Completed
	OVX836	Subunit		NCT03594890	1	Osivax	Completed
	BW-1014	Subunit	Nanoemulsion Adjuvanted	NCT05397119	1	BlueWillow Biologics	Recruiting
	Trivalent inactivated influenza virus vaccine Flucelvax(R) BPL-1357	Inactivated	Type 1 interferon	NCT00436046 NCT03845231 NCT05027932	1/2	NIAID	Completed
	NB-1008	Inactivated	W805EC	NCT01354379 NCT01333462	1	NanoBio Corporation	Completed
	H5N1 Influenza Vietnam 1194 Hemagglutinin (HA)	Adenoviral-vectored		NCT01806909	1	NIAID	Completed
	Hemagglutinin (HA)	Inactivated	DCB07010	NCT03293732	1	Advagene Biopharma Co., Ltd.	Completed
	AD07030	Inactivated	LTh(αK)	NCT03784885	2	Advagene Biopharma Co., Ltd.	Completed
	GelVac™ nasal powder H5N1	Inactivated H5N1		NCT01258062	1	Ology Bioservices	Completed
HIV	EN41-FPA2 HIV			NCT01509144	1	PX’Therapeutics	Completed
	Ad4-mgag Ad4-Env150KN	Ad4-based		NCT01989533 NCT03878121	1	NIAID	Completed/Recruiting
	Human Immunodeficiency Virus glycoprotein 140 (vaccine)	Glycoprotein 140	LTK63	NCT00369031	1	St George’s, University of London	Terminated
	Vacc-4x	Peptides	Endocine	NCT01473810	1	Oslo University Hospital	Completed
	MYM-V101	Subunit	virosome	NCT01084343	1	Mymetics Corporation	Completed
COVID-19	Gam-COVID-Vac	Adenoviral- vectored		NCT05248373	1/2	Gamaleya Research Institute of Epidemiology and Microbiology	Not yet recruiting
	AZD1222	Adenoviral- vectored		NCT05007275	1	Imperial College London	Recruiting
	AD17002	Subunit	LTh(αK)	NCT05069610	1	Advagene Biopharma Co., Ltd.	Recruiting
	MV-014-212	RSV-vectored		NCT04798001	1	Meissa Vaccines, Inc.	Recruiting
	DelNS1-nCoV-RBD LAIV	Influenza virus Vectored		NCT04809389	1	The University of Hong Kong	Active, not recruiting
	DelNS1-2019-nCoV-RBD-OPT1	Influenza virus Vectored		NCT05200741	1	The University of Hong Kong	Not yet recruiting
	ACM-001	Subunit	CpG	NCT05385991	1	ACM Biolabs	Not yet recruiting
	SC-Ad6-1	Ad6-vectored		NCT04839042	1	Tetherex Pharmaceuticals Corporation	Recruiting
	AVX/COVID-12 Vaccine	NDV Vectored		NCT05205746	2	Laboratorio Avi-Mex, S.A. de C.V.	Recruiting
	NDV-HXP-S	NDV Vectored		NCT05181709	1	Sean Liu	Recruiting
RSV	SeVRSV	Sendai Virus Vectored		NCT03473002	1	NIAID	Completed
	Ad26.RSV.preF	Ad26-vectored		NCT03334695	2	Janssen Vaccines and Prevention B.V.	Completed
	RSV001	PanAd3-RSV		NCT01805921	1	ReiThera Srl	Completed
Ebolavirus Zaire Glycoprotein	HPIV3/ΔHNF/EbovZ GP	HPIV3-vectored		NCT03462004	1	NIAID	Suspended
Allergen	CYT005-AllQbG10		Immunomodulatory principle (QbG10)	NCT00293904	1	Cytos Biotechnology AG	Completed
Anthrax	BW-1010	Recombinant Anthrax Vaccine	Nanoemulsion Adjuvanted	NCT04148118	1	BlueWillow Biologics	Completed
Norwalk	Norwalk VLP Vaccine		VLP	NCT00806962	1	LigoCyte Pharmaceuticals, Inc.	Completed
Cholera	Cholera Toxin B Subunit (CTB)	Subunit		NCT00820144	1	Centre Hospitalier Universitaire de Nice	Completed
Tuberculosis	Ag85B-ESAT6 fusion protein H1	Subunit	LTK63	NCT00440544	1	St George’s, University of London	Terminated

## Data Availability

Not applicable.

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
