# Peer review of "Development of Nasal Vaccines and the Associated Challenges"

_pharmaceutics, 2022, doi:10.3390/pharmaceutics14101983_

Round 1
Reviewer 1 Report
The review article entitled “Development of nasal vaccines and the associated challenges” proposes to describe the most recent strategies used for developing nasal vaccines. It brings nice explanation about the nasal structures and immune responses, examples of nasal vaccines, already licensed or in clinical development, challenges and some solution for nasal immunization. The text is well written and an important contribution to the field, but some points could be clarified to improve the text.
To increase the understanding of a more general audience, I would suggest including identification of cell types in Figure 1 B. For example, are dendritic cells represented in green? Which one are the M cells? The figure 1 B shows yellow, blue and purple cells, which cells are represented by each color? Different kind of antibodies are also shown, but no indication about which one is IgA, IgG or other type. The lymphoid follicle should also be indicated the figure.
Along the text, several examples are given, but it does not become clear if there are strategies that are more appropriated to certain types of antigens than others, for instance, which delivery system would be better for nucleic acid, or for proteins. It is also not clear if adjuvants and/or delivery systems should also be employed for live attenuated bacteria. Moreover, it would be very helpful for the readers if the authors could give some clues on which combination of antigen and adjuvant would give the best immune response, including a table or figure representing the most favorable combinations.
The text does not make it clear if it will be possible to formulate other types of nasal vaccines that were not based live attenuated microorganisms or viral vectors (item 3). Although some challenges are described in item 4, it is not clear, at least for this reader, if overcoming those challenges would be enough to the success of subunit, nucleic acid or inactivate nasal vaccines.
Chitosan and derivatives have also mucoadhesive properties (item 4), but their properties are discussed only in the item 7.1.5. In benefit of consistency, chitosan should be mentioned also in the item 4.
Polymeric nanoparticles were not included in Figure 2. Why?
The main concern on toxoid adjuvants administered via the nasal route is the risk of damage the olfactory nerve. As far as I know, the perception of this risk was not eliminated even when using detoxified versions of CT and LT. The authors could include some comments about how to overcome the resistance to employing toxoids as adjuvants for nasal vaccines.
L. 8, which other pathogenic microorganisms could infect host besides viruses, bacteria and fungi?
L. 29, revise punctuation.
L. 66, “illustrated in Figure 1” would be better than “just as Figure 1”
L. 154, nasal immunization with Bacillus Calmette-Guérin (BCG) vaccine does not serve for the treatment of Mycobacterium tuberculosis, the cited article proposes that nasal administration of BCG improves the protection against tuberculosis.
L. 475, please separate “trialon”.
L. 595, please separate “trialof”
Author Response
The review article entitled “Development of nasal vaccines and the associated challenges” proposes to describe the most recent strategies used for developing nasal vaccines. It brings nice explanation about the nasal structures and immune responses, examples of nasal vaccines, already licensed or in clinical development, challenges and some solution for nasal immunization. The text is well written and an important contribution to the field, but some points could be clarified to improve the text.
To increase the understanding of a more general audience, I would suggest including identification of cell types in Figure 1 B. For example, are dendritic cells represented in green? Which one are the M cells? The figure 1 B shows yellow, blue and purple cells, which cells are represented by each color? Different kind of antibodies are also shown, but no indication about which one is IgA, IgG or other type. The lymphoid follicle should also be indicated the figure.
Answer:Thanks for the advice, the Fig 1 has been revised with identification of cell types(different color has also been marked )
Along the text, several examples are given, but it does not become clear if there are strategies that are more appropriated to certain types of antigens than others, for instance, which delivery system would be better for nucleic acid, or for proteins. It is also not clear if adjuvants and/or delivery systems should also be employed for live attenuated bacteria. Moreover, it would be very helpful for the readers if the authors could give some clues on which combination of antigen and adjuvant would give the best immune response, including a table or figure representing the most favorable combinations.
Answer:Thanks for the advice, we also wants to find the best strategies for the appropriated to certain types adjuvant suitible for certain types of antigens, liposome and chitosan could be used for subunit, inactivated or nucleic acids vaccine, which has been emphasized in the article. The other adjuvant mainly for the subunit, inactivated or peptides vaccine.
Live attenuated bacteria nasal vaccine also is under development, we added this part in the article (lines 234-240), but we did not found any adjuvants and/or delivery systems used for live attenuated bacteria nasal vaccine, so no relevant sentence or reference is provided in the article.
In fact, various laboratories have described the certain feasibility of various adjuvants used for nasal vaccine, it is difficult to describe which vaccines and adjuvants combination could give the best immune response, according to the related research results, the combination of immunostimulatory adjuvants and/or delivery systems provide the satisfactory result, but the safety is bothersome, We provided the progress of clinical trials of peptide, subunit, inactivated, and viral vector-based nasal vaccines, which could provide the potentially successful vaccine and antigen combinations. In the discussion, we describe the prospects for nasal mucosal adjuvants
The text does not make it clear if it will be possible to formulate other types of nasal vaccines that were not based live attenuated microorganisms or viral vectors (item 3). Although some challenges are described in item 4, it is not clear, at least for this reader, if overcoming those challenges would be enough to the success of subunit, nucleic acid or inactivate nasal vaccines.
Answer: Thanks for the advice, The item 3 “ the insight into the available licensed influenza nasal vaccine”, which told us that live attenuated nasal vaccines are possible for development, we also give the example SARS-CoV-2 vaccine under development for the nasal vaccine of subunit, nucleic acid or inactivate nasal vaccines (Nine SARS-CoV-2 nasal vaccines, including the inactivated, live attenuated, protein subunit, nucleic acid, viral vector, and other vaccines, are currently in clinical trials, which further corroborates the feasibility of nasal vaccines [44].
The challenge of attenuated vaccines is mainly virulence, but for the other type vaccine, which should be consider for the item 4 challenges. Line 149-150.
it is not clear, at least for this reader, if overcoming those challenges would be enough to the success of subunit, nucleic acid or inactivate nasal vaccines.
Answer: As no successful nasal mucosal vaccine besides influenza(LAIV), all the adjuvants or delivery system sated in this paper for nasal vaccine are under trial, so we revised and declared that: Some adjuvant-added nasal vaccines comprise both immune-enhancing and delivery-facilitating adjuvants to promote the delivery of vaccines to immune cells and also enhance the immunogenicity of vaccines. The establishment of novel adjuvanted nasal vaccines as safe and effective requires validation through extensive preclinical and clinical trials, and athough the successful development of nasal vaccines has a long way to go, it owns positive implications, if the safety of the adjuvant-added nasal vaccines guaranteed, the effective candidate for subunit, nucleic acid or inactivate nasal vaccines would be enough to the success.(lines 622-632)
Chitosan and derivatives have also mucoadhesive properties (item 4), but their properties are discussed only in the item 7.1.5. In benefit of consistency, chitosan should be mentioned also in the item 4.
Anwer: In the item 4, the chitosan was stated as “Different antigen formulations have been developed, such as mucoadhesives, to reduce the effects of nasal mucosal cilia on antigen clearance[49, 50]”. the reference 49 mentained the N-trimethyl chitosan.
For better understanding this sentence has been revised as “Different antigen formulations have been developed, such as mucoadhesives (chitosan), to reduce the effects of nasal mucosal cilia on antigen clearance [49, 50](lines 161-162)
Polymeric nanoparticles were not included in Figure 2. Why?
Answer: The Polymeric nanoparticles is a relatively large categoryconsider including ISCOMs, nanoemulsion and so on, which is contained in Fig 2
The main concern on toxoid adjuvants administered via the nasal route is the risk of damage the olfactory nerve. As far as I know, the perception of this risk was not eliminated even when using detoxified versions of CT and LT. The authors could include some comments about how to overcome the resistance to employing toxoids as adjuvants for nasal vaccines.
Answer: Thanks for the advice, this has been added (lines 313-318)
- 8, which other pathogenic microorganisms could infect host besides viruses, bacteria and fungi?
Answer: for example: Mycoplasma, Chlamydia,Toxoplasma dondii, Crytosporidium parvum
L29, revise punctuation.
Answer:Thanks for the advice, this has been revised (line 32)
- 66, “illustrated in Figure 1” would be better than “just as Figure 1”
Answer:Thanks for the advice, this has been revised (line 65)
- 154, nasal immunization with Bacillus Calmette-Guérin (BCG) vaccine does not serve for the treatment of Mycobacterium tuberculosis, the cited article proposes that nasal administration of BCG improves the protection against tuberculosis.
Answer:In our article, the sentence“Promising preclinical data has been generated for nasal immunization with Bacillus Calmette-Guérin (BCG) vaccine for the treatment of Mycobacterium tuberculosis”means nasal administration of BCG improves the protection against tuberculosis just as the cited article stated(line
- 475, please separate “trialon”.
Answer:Thanks for the advice, this has been revised (lin449)
- 595, please separate “trialof”
Answer:Thanks for the advice, this has been revised (line 559)

Reviewer 2 Report
Very nice of review article on nasal vaccines. It is well investigated of previous and on-going vaccine development for infection with tables and figures for the readers to understand present and future trials of them.
A couple of suggestions is written in the following to revise your draft to tune up a little more.
1. relative pronouns used very often, but long sentences could be simplified for the readers to better understand, so you may ask native speaker to revise sentences through out.
2. in page 2, at nasal structure and immune responses, line 70, you are saying that NALT(nasal associated lymphoid tissue) constitutes the Waldeyer ring in humans, in accordance with two references 11,12, but in a clinical situations, we have still a discussion how we consider details of tonsillar tissues inhuman as responding to NALT in rodents. So, I recommend you to express like as "Human Waldeyer ring tonsillar tissues can be considered to correspond to the NALT in rodents."
3. IF you prefer it, the concept of compartmentalization on induction and effector site could be added in introduction or somewhere.
Author Response
Very nice of review article on nasal vaccines. It is well investigated of previous and on-going vaccine development for infection with tables and figures for the readers to understand present and future trials of them.
A couple of suggestions is written in the following to revise your draft to tune up a little more.
- relative pronouns used very often, but long sentences could be simplified for the readers to better understand, so you may ask native speaker to revise sentences through out.
Answer: Thanks for the advise, the paper has been revised by native speaker.
- in page 2, at nasal structure and immune responses, line 70, you are saying that NALT(nasal associated lymphoid tissue) constitutes the Waldeyer ring in humans, in accordance with two references 11,12, but in a clinical situations, we have still a discussion how we consider details of tonsillar tissues inhuman as responding to NALT in rodents. So, I recommend you to express like as "Human Waldeyer ring tonsillar tissues can be considered to correspond to the NALT in rodents."
Answer: Thanks for the advise, This sentence has been revised(lines 69-70).
- IF you prefer it, the concept of compartmentalization on induction and effector site could be added in introduction or somewhere.
Answer:The article stated that: An inductive site is where the induction of the immune response begins. This is the site where the antigen first reaches the respiratory tract, then crosses the mucus layer, and is finally absorbed. The inductive site mainly comprises mucosal lymphoid follicles located within the respiratory zone, which are collectively referred to as the nasal-associated lymphoid tissue (NALT) (lines 65-67)
An effector site in the nasal mucosal immune system is where the immune responses of B cells and T cells occur. The effector sites include the lamina propria (LP) and the intraepithelial layer of the respiratory mucosa(lines 85-87)
